# Residential Mobility of a Cohort of Homeless People in Times of Crisis: COVID-19 Pandemic in a European Metropolis

**DOI:** 10.3390/ijerph19053129

**Published:** 2022-03-07

**Authors:** Agathe Allibert, Aurélie Tinland, Jordi Landier, Sandrine Loubière, Jean Gaudart, Marine Mosnier, Cyril Farnarier, Pascal Auquier, Emilie Mosnier

**Affiliations:** 1Department of Psychiatry, Assistance Publique—Hôpitaux de Marseille, 13385 Marseille, France; agathe.allibert@gmail.com (A.A.); aurelie.tinland@gmail.com (A.T.); 2Epidemiology of Zoonoses and Public Health Research Unit (GREZOSP), Faculty of Veterinary Medicine, Université de Montréal, Saint-Hyacinthe, QC J2S 2M2, Canada; 3EA 3279: CEReSS—Health Service Research and Quality of Life Center, School of Medicine—La Timone Medical Campus, Aix-Marseille University, 13005 Marseille, France; sandrine.loubiere@univ-amu.fr (S.L.); pascal.auquier@univ-amu.fr (P.A.); 4Aix Marseille Univ, IRD, INSERM, SESSTIM, ISSPAM, AP-HM, La Timone Hospital, BioSTIC, Biostatistic & ICT, 13005 Marseille, France; jordi.landier@ird.fr (J.L.); jean.gaudart@univ-amu.fr (J.G.); 5Support Unit for Clinical Research and Economic Evaluation, Assistance Publique Hôpitaux de Marseille, 13385 Marseille, France; 6Médecins du Monde—Doctors of the World, 13003 Marseille, France; marinemosnier@gmail.com; 7Laboratoire de Sciences Sociales Appliquées/Projet ASSAb, 13001 Marseille, France; cyril.farnarier@gmail.com

**Keywords:** SARS-CoV2, COVID-19, homeless people, public health, vulnerable population, seroprevalence, cohort, residential mobility

## Abstract

Most vulnerable individuals are particularly affected by the COVID-19 pandemic. This study takes place in a large city in France. The aim of this study is to describe the mobility of the homeless population at the beginning of the health crisis and to analyze its impact in terms of COVID-19 prevalence. From June to August 2020 and September to December 2020, 1272 homeless people were invited to be tested for SARS-CoV-2 antibodies and virus and complete questionnaires. Our data show that homeless populations are sociologically different depending on where they live. We show that people that were living on the street were most likely to be relocated to emergency shelters than other inhabitants. Some neighborhoods are points of attraction for homeless people in the city while others emptied during the health crisis, which had consequences for virus circulation. People with a greater number of different dwellings reported became more infected. This first study of the mobility and epidemiology of homeless people in the time of the pandemic provides unique information about mobility mapping, sociological factors of this mobility, mobility at different scales, and epidemiological consequences. We suggest that homeless policies need to be radically transformed since the actual model exposes people to infection in emergency.

## 1. Introduction

The mobility of the homeless within large North American and European cities prior to the health crisis remains a little studied phenomenon [1]. Some studies indicate that this mobility is greater than that of the general population in the same city [2,3]. Homeless people move within a city in order to find sustenance. According to Kaufman [4], research concerning the mobility of homeless people emphasizes moves within cities and reveals seven factors that are worthy of note: housing; labor markets; social, health, and justice services; personal health; the attributes of different places; interpersonal networks; and how mobility is socially differentiated. Homeless people from all kinds of accommodation were found to have a notable daily mobility [5], but little is known about their residential mobility over several months. Homeless people may also settle transiently, episodically, or chronically in a shelter [6].

To help homeless people exit homelessness, authorities implement rehousing programs. Most traditional rehousing programs assume that a person’s homelessness is the result of poor decision-making [7,8]. Most traditional programs such as “step-by-step” are clustered and supervized so that residents can become “housing ready” [9]. Although the goal of rehousing programs is to end homelessness, on average it takes over 10 years for people to move out of these types of living conditions [10]. The reality is that many people are caught in an “institutional circuit” of homelessness, hospitals, and prisons [11]. There is growing evidence that, while traditional housing programs manage homelessness, they are not particularly effective in ending it [12,13].

On 11 March 2020, the COVID-19 epidemic was declared a pandemic by the WHO [14]. The consequences of this crisis are also economic and social, particularly affecting the most vulnerable people [3,15]. Studies have shown that homeless people are at a greater risk of SARS-CoV-2 infection than the general population [16,17]. Homeless people have suffered from the disruption of their living and collecting places due to the epidemic. Allaria et al. (2021) [18] reported that the lockdown of the general population in France severely impacted the survival systems of the populations furthest from housing, with alarming rates of people without access to water or food. In addition, 77% of homeless participants reported that they encountered significant financial difficulties. Under the effects of a pandemic, there are additional constraints that are specific to the health crisis, which compound those constraints that are specific to homelessness: emergency accommodation link, continuation of a disrupted economic activity, etc. NGOs and the French public authorities took measures to help homeless people and provide them with shelter, especially during the initial lockdown. Conversely, emerging data have shown that homeless people living on the street appear to be at a lower risk of SARS-CoV-2 infection than people that are living in shelters [19,20]. This observation may be due to asymptomatic infections, which account for approximately 17% of cases [21]. The problem of asymptomatic infection is particularly important in congregate shelters, as asymptomatically infected persons can unknowingly transmit the infection to a large number of people in a short period of time [21]. Disrupted mobility may also play a role in the patterns of SARS-CoV-2 infection in the homeless population. Indeed, shelters where large numbers of people transit are more likely to have a high prevalence of SARS-CoV-2 infection [19,20].

A study of homeless people in the Netherlands indicates that mobility and access to shelters that were opened at the beginning of the pandemic differed according to the category of origin of precarious people [22]. However, we do not know what socio-economic variables are associated with them and what the exact consequences are for the mobility and sheltering of these people.

The pandemic is a good example of the effectiveness of the traditional type of rehousing program in times of crisis. It allows us to see how the pandemic accelerated the problems of access to housing in various vulnerable populations.

We hypothesized that the mobility of homeless people would be disrupted by the SARS-CoV-2 epidemic in terms of housing type and between neighborhoods in the city. We hypothesized that this mobility would differ according to the socio-economic characteristics of the homeless and that this mobility would have an impact in terms of epidemiology.

To study this hypothesis, we followed a large cohort of the most precarious people in Marseille, a large European city with a high rate of poverty and inequality. Marseille is a city of a rich country, a gateway to France and the European Union for many people from poorer countries. It could be representative of a city in a rich country that is close to poor countries attracting a high level of illegal and legal immigration. The fact that this city has many well identified associations allowed us to have a large number of respondents to our study, which provide a good representation of the precarious populations we studied.

In this study, we aimed to assess the homeless mobility and its epidemiological consequences in the context of the COVID-19 crisis.

## 2. Materials and Methods

### 2.1. Study Design

We conducted a prospective population-based cohort study of homeless people that were living on the streets, in shelters, or squats and slums: the COVID-Homeless survey (registered on ClinicalTrials, NCT04408131, 29 May 2020). This study aimed to exhaustively include participants from all shelters and outreach teams of the city. Each subject was tested twice: the first study lasted from 5 June–5 August 2020 (first campaign), and the second three months later, 11 September–18 December 2020 (second campaign). The homeless persons that were followed were tested for SARS-CoV-2 antibodies and answered a questionnaire concerning their life habits, socio-demographic data, and recent geographic and residential movements.

### 2.2. Study Area

The study area was the city of Marseille. Marseille is the second largest city in France, but also the poorest. It is situated in the Southeast of France, in the Bouche du Rhône department, which was particularly affected by SARS-CoV-2. A large public health survey estimated the seroprevalence of SARS-CoV-2, based on 12,400 samples that were taken in May 2020, to be 4.5% for the whole of France and 5.2% for the French region of Provence-Alpes-Côte d’Azur in which our study area was located [23]. On 17 March 2020, France entered its first lockdown, which ended on 11 May 2020. Following a resurgence of the epidemic after the summer of 2020, a second national confinement was decreed from 30 October to 15 December 2020. Marseille, similar to all French cities, is divided into 3 administrative divisions, from the largest to the smallest: 16 districts, 111 neighborhoods, and 742 units of equal size, called IRIS [24]. Most statistics and maps in this study are at the neighborhood scale, such as Figure 1, which depicts a map of the districts and neighborhoods in Marseille, France. This map shows the study area and is useful for associating district names with their locations in spatial analysis results.

Marseille is the second most populous city in France, suffering a high level of poverty [25]. More than one out of two residents live below the poverty line (51.3%) [26]. Marseille’s impoverished neighborhoods contrast markedly with the wealthy areas of the city, which benefit from good access to personal services, health institutions, and shops, demonstrated by INSEE (French National Institute of Statistics and Economic Studies) classifications. Figure 2 shows the heterogeneous distribution of emergency accommodation in Marseille in relation to the different types of services. Figure 2, in the context of a study on precariousness and mobility, presents the state of healthcare supply and potentially the geographical factors of health inequality.

### 2.3. Population

In order to focus on the homeless people the furthest from housing, we decided to select those characterized by the greatest residential instability: people sleeping rough, in squats or slums, in stabilization shelters, in emergency shelters, or hostels, respectively, corresponding to the following categories of the European typology of homelessness (ETHOS): ETHOS 1, 2, 3, and 8 [27]. In the absence of a point-in-time count, random sampling was impossible. 

However, data from the local orientation system for emergency and transitional accommodation (SIAO) and the NGO Doctors of the World estimated that in 2020, at the beginning of the COVID-19 outbreak, there were 2322 homeless adults living in emergency, transitional shelters, or hostels and 619 to 817 living in squats or slums. No point-in-time census was available for people that were living on the streets in Marseille. 

We set a 2-month inclusion period, during which we systematically offered all homeless people that were aged over 18 to participate in the study. The recruitment of participants was also facilitated by the “Accés aux Soins des Sans Abris (ASSAb) network” of assistance to homeless people: 18 homeless outreach teams that were working in streets, hotels, squats, or slums; 5 emergency shelters; and 10 transitional accommodations. All the participants provided written informed consent. ETHOS categories were allocated according to the primary living location for the people that were questioned.

### 2.4. Study Design

The participants were interviewed by trained cultural community health workers in the language of the participants and also using a questionnaire. This questionnaire collected data on the sociodemographic characteristics, medical history and type, and history of housing. The homes of the participants were georeferenced. Questions were asked by trained local interviewers in the participants’ native language to improve comprehension and to minimize information bias [28].

### 2.5. Biological Analysis

All the participants had a rapid serological test. People with symptoms were invited to be tested by SARS-CoV-2 PCR screening. We used the rapid serological test “Biosynex COVID-19 BSS^®^”, providing the information about the presence of immunoglobulins M (IgM) and G (IgG) in 10 min. A Biosynex vitaPCR^®^ was performed in case of symptoms of COVID-19 disease during the interviews [29], which provides results within 20 min.

### 2.6. Outcomes and Data Analysis

SARS-CoV-2 history of infection was defined by a positive SARS-CoV-2 serology (IgM or IgG) during the study period. All of the statistical analyses were carried out using R software [30], and differences with *p* values of <0.05 were considered statistically significant.

Maps were made using QGIS software [31]. Data on the administrative boundaries of the city come from French government databases. In the absence of indications to the contrary, complete case analyses were performed.

#### 2.6.1. Socio-Demographic Factors and Living Areas

A Hill and Smith analysis were performed with the R package ade4 [32,33]. This analysis generalizes the PCA (principal components analysis) method to be used with quantitative variables and factors [34]. The results and graphs read like those of a PCA [35]. This analysis was based on the responses of the participants in the first testing session. We used the sociodemographic characteristics variables of our population to perform this analysis. We used stochastic regression imputation to assess the variables for individuals with missing data, using the R package ‘mice’ [36].

#### 2.6.2. Relation between Mobility at the Individual Scale and Infection with SARS-CoV-2

To find out if the number of accommodations in the past year was significantly associated with having a positive serological test for SARS-CoV-2, we used a univariate logistic regression model. The independent variable was the number of accommodations since the pandemic and the dependent variable was the presence or absence of antibodies to SARS-CoV-2 at the individual level. To know the number of accommodations since the pandemic, we used the following question: “What is the person’s current housing?” followed by the question “and before?”. We repeated the “and before?” question until we went back to the beginning of the pandemic. The total number of different housing units that were filled in corresponded to the number of accommodations since the pandemic.

#### 2.6.3. Life Paths: Mobility at the Housing Scale

To illustrate mobility at the housing level, we used a Sankey diagram that shows mobility between ETHOS at 5 different time periods. The different periods were as follows: before the beginning of the health crisis (24 January 2020), before the lockdown (between 24 January and 16 March 2020), during the lockdown (between 16 March and 11 May 2020), and after the lockdown (between 11 May 2020 and 5 August 2020). All of this information was requested during the first campaign session. We also collected this information during the second test session (between 11 September and 18 December 2020) (second campaign). A Sankey Diagram was made using R software and the package networkD3 [37].

#### 2.6.4. Mobility and Spatial Epidemiology at the Neighborhood Scale

Satscan software [38] was used for cluster analysis to detect the possible locations where the number of cases was higher than expected. We performed cluster analysis for the serological results for the first and second campaigns. We purely used spatial analysis for scanning for clusters with high rates. We used the Bernoulli distribution and an elliptic window shape for scanning, with a maximum spatial cluster size of 50 percent of population at risk.

### 2.7. Ethical Approval

The study was approved by the ethics committee Comité de Protection des Personnes d’Ile de France VI on 28 May 2020 (number 44–20). All of the people that were included in this study provided written informed consent. The database was anonymized and declared to the French regulatory commission (Commission Nationale Informatique et Libertés, CNIL, n°2018172v0).

## 3. Results

We included 1272 people in the cohort (Table 1) and 738 provided additional data during the second serological testing step (58.02% of included people).

In the first campaign, the majority of the individuals were male (70.29%, 894/1272), with an average age of 40.06 years (standard error: 0.40) and 6.01% (74/1231) testing positive for SARS-CoV-2. In the second campaign, the majority of the individuals were male (71.7%, 545/738), with an average age of 41.76 years (standard error: 0.54). 18.86% (136/721) had SARS-CoV-2 antibodies.

### 3.1. Socio-Demographic Factors and Living Areas

In the Hill and Smith analysis, axis 1 contrasted two types of people. The first group comprised of people that were born in France, who take drugs, whose education was lower secondary, who were isolated parents, and live in ETHOS 1. The opposing group characteristics were female, that were born in European countries including non-members of the European Union (EU), who lived in families, and lived in ETHOS 8 housing (Figure 3, Appendix A Table A1). Axis 2 opposed two types of people. The first group concerned people that were born in countries of sub-Saharan or Southern African countries, Middle-Eastern countries, and North and South American countries, who did not smoke. They were contrasted with people that were born in European Union countries and in France, who took drugs, who have been homeless for more than 5 years, and lived in ETHOS 1 housing (Figure 3, Appendix A Table A1). The housing situation was an important variable in this analysis (Appendix A Table A1). On the first axis of the analysis, ETHOS 1, 2, and 3 are opposed to ETHOS 8. On the second axis, ETHOS 1 and 8 are opposed to ETHOS 3 and 2. 

### 3.2. Relation between Mobility at the Individual Scale and Infection with SARS-CoV-2

The average number of different accommodations since the pandemic was 1.718 (standard error: 0.034), with 94 missing data. The number of different accommodations since the pandemic was significantly associated with having a positive serological test for SARS-CoV-2 (Table 2).

### 3.3. Life Paths: Mobility at the Housing Scale

Before the beginning of the health crisis (January 2020), 13.08% of the people were counted in ETHOS 1 (166/1270), 35.2% in ETHOS 2 (447/1270), 38.19% (485/1270) in ETHOS 8, and 13.54% in ETHOS 3 (172/1270) (Figure 4). Between January 2020 and March 2020, beginning of the first lockdown, 13.63% (165/1211) of the population changed their accommodation status. During the first lockdown (March to May 2020), 15.27% (178/1166) of people moved. The most important flows were those of people going to ETHOS 2 (emergency shelters). Thus 30.56% (44/144) of people in ETHOS 1 before the first lockdown went to ETHOS 2 during the first lockdown, 9.84% (44/447) of people in ETHOS 8 went to Ethos 2, and 27.27% (12/44) of people in the ‘other’ category also went to ETHOS 2. Although a number of people left ETHOS 2 to go primarily to ETHOS 3 (4%, 14/352) between these dates, the flows were positive for ETHOS 2, which saw its population increase from 29.10% (353/1213) of reported housing types to 36% (440/1223) during the lockdown. After the first lockdown (ending in 11 May 2020), 13.85% (168/1213) of people moved. ETHOS 2 continued to receive people. Thus 31% (31/100) of people that were in ETHOS 1 during the lockdown went to ETHOS 2 after the lockdown. 9.18% (37/403) of people in ETHOS 8 went to ETHOS 2 after the lockdown, and 27% (10/36) of people in the other category went to ETHOS 2 as well. Between the two testing sessions (May to December 2020), 23.17% (165/712) of people moved. The most important flow was between people in ETHOS 2 after the lockdown and those in ETHOS 3 during the second testing session: 24.4% (71/291). This flow corresponded to people in emergency shelters who went to homeless hostels (transitional hostels, temporary accommodation, or transitional accommodation with support).

### 3.4. Mobility and Spatial Epidemiology at the Neighborhood Scale

For the population dynamics of mobility between the first and second campaigns, we have information about 377 people in the first campaign and 721 in the second campaign. We have information about the population dynamics in 45 of the 110 neighborhoods in Marseille. Of these 45 neighborhoods, 21 (46.7%) lost people between the first campaign and the second campaign, 19 (42.22%) gained people, and 5 (11.11%) had an equivalent number of respondents (Figure 5).

For the first period of testing (from 5 June to 5 August 2020), we had the test results of 377 people with associated geographical coordinates. We tested 39 neighborhoods out of the 110 in the city of. The prevalence per neighborhood was between 0 and 0.5 (Figure 6, Appendix B Table A2). The total prevalence, across all the neighborhoods combined (for the 377 people) was 2.65% (IC95%: 1.03–4.27%).

For the first campaign, we identified a non-significant cluster in the neighborhoods north-west of Marseille (population = 168, Number of cases = 8, expected cases: 4.46, observed/expected: 1.80, relative risk: 4.98, log likelihood ratio: 2.711407, *p*-value: 0.75, not a Gini Cluster) (Figure 7, Appendix B Table A2).

For the second period of testing (from 11 September to 18 December 2020), we had the tests of 721 people with associated geographical coordinates. We tested 43 neighborhoods out of the 110 in the city of Marseille. The prevalence per neighborhood was between 1 and 0.024 (Appendix B Table A3, Figure 8). The total prevalence, across all the neighborhoods combined (for the 721 people) was 10.12% (IC95%: 7.923–12.23).

For the period of the second campaign, four clusters were identified, two of which were significant and two not significant (Table 3, Appendix B Table A3, Figure 9): a significant cluster of 6 neighborhoods (cluster 1), around the old port of Marseille; a significant cluster of 16 neighborhoods (cluster 2), located in the center of Marseille; a non-significant cluster of 8 neighborhoods (cluster 3) in the north of Marseille; and a non-significant cluster which was located in the neighborhood of La Villette (cluster 4).

## 4. Discussion

The interest of this article is two-fold: it presents both unique data on homeless characteristics, mobilities, and their consequences in times of crisis in a large European city. We have shown that the homeless population in a large European city, such as Marseille, is very heterogeneous, both in terms of personal circumstances and type of homelessness (on the street, in emergency accommodation, etc.). Our study population was mobile in different ways during the year 2020, corresponding to the beginning of the COVID-19 crisis in France. Mobility varied according to the type of homelessness that was experienced at the beginning of the crisis. We have shown that mobility within the different neighborhoods of the city probably explains the evolution of the cluster locations as the epidemic progressed. We also showed a positive association between a large number of housing changes the probability of having anti-SARS-CoV-2 antibodies. We also report that there is an association between a panel of socio-economic variables and the type of housing that people had before the epidemic. We then show that the rate of crisis-related rehousing in emergency shelters differed according to the type of housing that people had before the epidemic. The classic emergency housing model, therefore, fails to provide shelter for some of the most precarious types of population: families, people living in slums, etc. 

We were able to include many people in our cohort and had a response rate of 58.02%. This inclusion is satisfactory given the type of people that were targeted and the health crisis that was unfolding at the time. This study was made possible by the close involvement of local NGOs in the field. We want to emphasize the need for close cooperation between researchers and NGOs in the field in order to reach the most vulnerable in studies of this magnitude. Compared to the general population of Marseille, the homeless population is younger and consists of more men; 47% of male in the general population in Marseille in 2018, according to [39].

Different types of homelessness had a clear relationship to personal characteristics. The country of birth was a significant variable in the analysis affecting the type of homelessness that people experienced. It is possible to distinguish several groups: one group was made up of people that were born in France, who consumed alcohol, tobacco, and other drugs whilst living on the streets. Another group comprised of people that were living with their families, in squats, and shantytowns, that were born in Europe (outside and inside the European Union), who tended to remain in one area. The last group, less differentiated, was made up of people that were living in emergency shelters and transit shelters, parents, or single adults, with a secondary level of education, that were born in Africa, the Middle East, or America (North and South). Whilst our study highlighted the heterogeneity of the homeless population within Marseille, we were able to draw attention to the existence of new categories of people and the need for help that is adapted to their specific needs. The homeless population is not homogeneous. In the rest of Europe, as in Marseille, the socio-demographic characteristics are multiple and include groups requiring specific and different care, such as migrants or people with psychological problems [40]. These needs require a better understanding of the different types of homelessness and the socio-economic factors that are associated with them. This study is the first to our knowledge that provides this information in a major European city. 

Our study showed that the flow of people between different types of accommodation increased from the lockdown onwards, compared to periods in between where the restrictions were eased. This was mainly because the flow of people that were living in squats and on the streets into emergency accommodation increased after the lockdown. Populations who were on the streets before the crisis were most likely to move to emergency accommodation. These observations reflect an effort by NGOs and politicians to encourage people into shelters during the first lockdown in France. This movement of people continued until the period from May to August 2020. During the second test session, the most notable population flows occurred from emergency shelters towards ETHOS 3 housing. This corresponds to a cessation of COVID-related emergency accommodation and to people moving to more stable shelters. According to the step-by-step model, this shift allowed for support of the people concerned and aimed at their insertion into housing. This was a standard institutional process, which accelerated after the COVID crisis following political commitments promising that people would not end up back on the streets.

Our observations question contemporary homelessness social policies. Our Sankey diagram showed that emergency accommodation is not limited to people staying under ‘emergency’ conditions. In an ideal world, the ‘step-by-step’ model aims to facilitate the progressive movement of homeless people from the streets to emergency shelters (ETHOS 2), and then onwards to stabilization shelters (ETHOS 3) in order to help prepare them for private housing. This model remains dominant at the policy level, despite the existence of other models such as housing first, which promotes direct and unconditional access to housing and has proven more effective at producing housing stability [41,42]. Our study shows (Figure 4) that there were very few instances of people moving from homeless directly to private housing, and lots of people in the step-by-step model, experiencing long stays in emergency shelters, and little access to stabilization shelters at a later stage. Furthermore, shelters seemed ill-adapted to families without education, which were staying in slums (Figure 3 and Figure 4). Although it seems that shelters appeared as the easiest solution to provide rapid protection for homeless people, there was no long-term solution after the initial emergency response. As discussed, these emergency solutions also involved greater risks of infection [15,43]. For the authors, the step-by-step model and emergency policies for homelessness need to be radically transformed. As has been demonstrated in other European countries on community integration, a more comprehensive and sustainable approach, such as housing first programs, should be promoted [44].

In this paper we have shown that the most mobile individuals, with a greater number of different dwellings in the past year, were at greater risk of testing positive for SARS-CoV-2. The association between mobility and infection has already been demonstrated in the general population [45,46], but this is the first time it has been observed using the number of dwellings as an indicator in the homeless population. This showed the importance of residential stability in order to comply with isolation and social distancing measures.

Our study also highlighted the mobility of the homeless population. People could be mobile within a neighborhood, a city, a country, or a continent (intra-European mobility, for example). We looked at mobility at the level of a city’s neighborhoods, the possible reasons for such mobility, and its consequences in epidemiological terms. In this part of the study, we observed an effective mobility between the first and second campaigns, with some neighborhoods gaining population, whilst others decreased. 

We have shown that the number of dwellings is associated with a higher probability of infection. Going to a shelter is also associated with an increased likelihood of infection [19,20], especially if the shelters were overcrowded or the sleeping spaces were shared with a large number of residents [19,20]. Both of these factors argue for greater stability of people in times of health crisis in non-overcrowded living spaces, and approaches such as housing first seem to be more appropriate to address these issues.

### Limitations

A selection bias cannot be ruled out since we had no reliable census data from which to perform random sampling. However, we aimed at exhaustiveness by systematically including all homeless adults that were encountered in the field during the inclusion period with our partners, which included all shelters and homeless mobile outreach teams in the city. This extensive recruitment and the overall size of our study population limits this bias. Homeless people living in the streets (ETHOS 1) were harder to reach and more are lost to follow-up, despite the committed involvement of all the study partners, including NGOs. Although some government measures increased the mobility of the homeless population from living on the streets to emergency accommodation, other measures had the potential to reduce population mobility. For example, the ‘winter eviction ban’, which forbids the eviction of a tenant during the winter months, was extended by decree (Ordinance n°2020-331 of 25 March 2000 (JORF 26 March 2000) [47]). Entry restrictions on homeless accommodation and restrictions on the length of stay in shelters was also suspended for the duration of the lockdown. The high levels of mobility that were observed in our study are perhaps surprising given this context and might lead one to expect higher mobility in this population today with the cessation of these measures. There was a bias concerning the map in Figure 5, as we had more spatial location information for people in the second campaign than in the first: the population dynamics of the neighborhoods, therefore, risks exaggeration towards the positive. Nevertheless, this map is still relevant for comparing the neighborhoods with each other. The lower number of spatialized data in the first campaign could also explain why the cluster that was identified there does not emerge as statically significant.

## 5. Conclusions

In summary, the mobility of homeless people at the city-scale is an important factor in better understanding the epidemiological dynamics for these populations. To date, these questions have been under-investigated, despite concerning the public health of the most precarious people. Herein this study, we highlight the need for further research on these important issues. These results encourage the implementation of management that is adapted to the specific situations of these particularly vulnerable populations in times of health crisis. As this crisis can be seen as an accelerator of the usual sheltering movements, this study also allows us to reconsider the rehousing program as a whole by advocating for housing first. This study is important at the city level for the adaptation of local strategies, but also at the European level for the implementation of more sustainable public housing programs.

## Figures and Tables

**Figure 1 ijerph-19-03129-f001:**
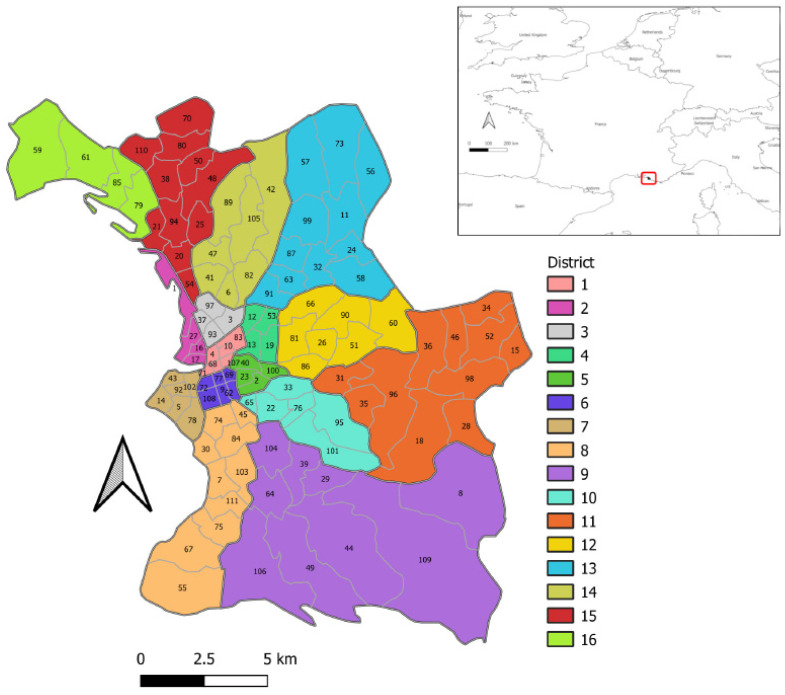
Map of the districts and neighborhoods in Marseille, France. The neighborhoods are in alphabetical order according to their name and names of districts are numbered.

**Figure 2 ijerph-19-03129-f002:**
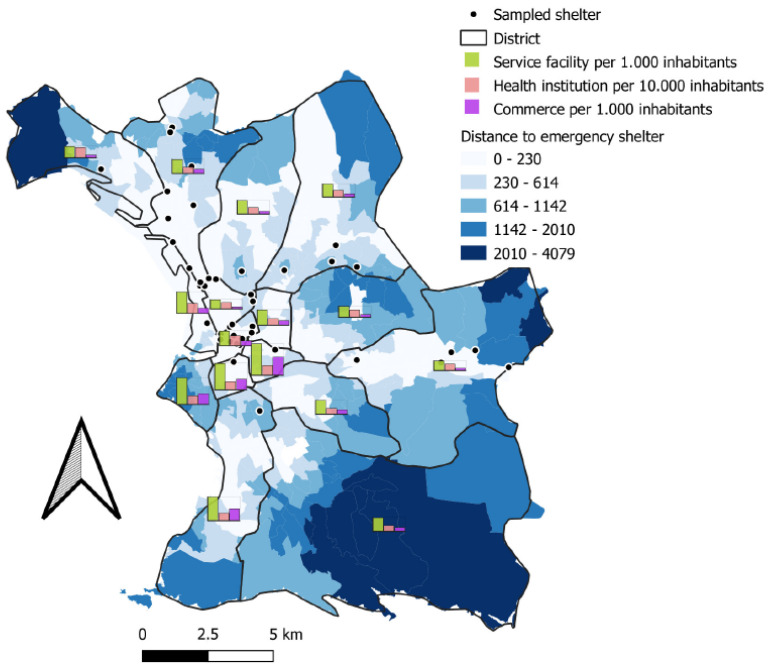
Map of the number of personal services (per 1000 inhabitants), health institutions (per 10,000 inhabitants), and commerce (per 1000 inhabitants) by district and distance to emergency shelters by IRIS, categorized into Jenks Natural Breaks Classification.

**Figure 3 ijerph-19-03129-f003:**
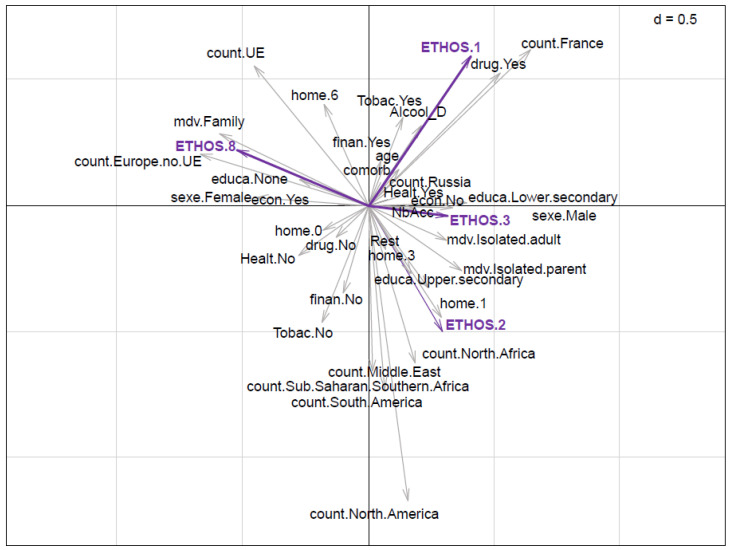
Scatter diagram of the projection of the variables in the first two axes of the Hill and Smith analysis with mixed quantitative variables and factors.

**Figure 4 ijerph-19-03129-f004:**
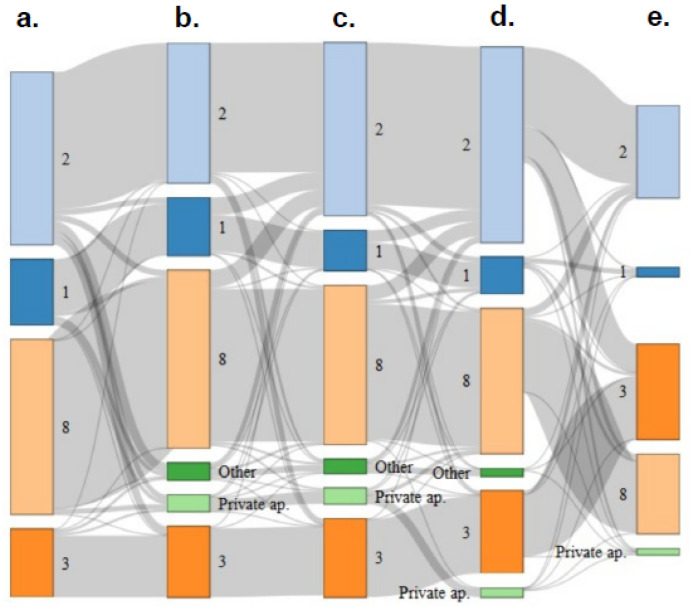
Sankey diagram of life paths in our study. Different periods are: (**a**) before the beginning of the health crisis (24 January 2020), (**b**) before the lockdown (between 24 January and 16 March 2020), (**c**) during the lockdown (between 16 March and 11 May 2020), (**d**) after the lockdown (between 11 May 2020, and 5 August 2020), and (**e**) during the second testing session (between 11 September and 18 December 2020). 1 indicates ETHOS 1, 2 indicates ETHOS 2, 3 indicates ETHOS 3, 8 indicates ETHOS 8, Private ap. indicate persons in private apartment and Other indicates other types of housing.

**Figure 5 ijerph-19-03129-f005:**
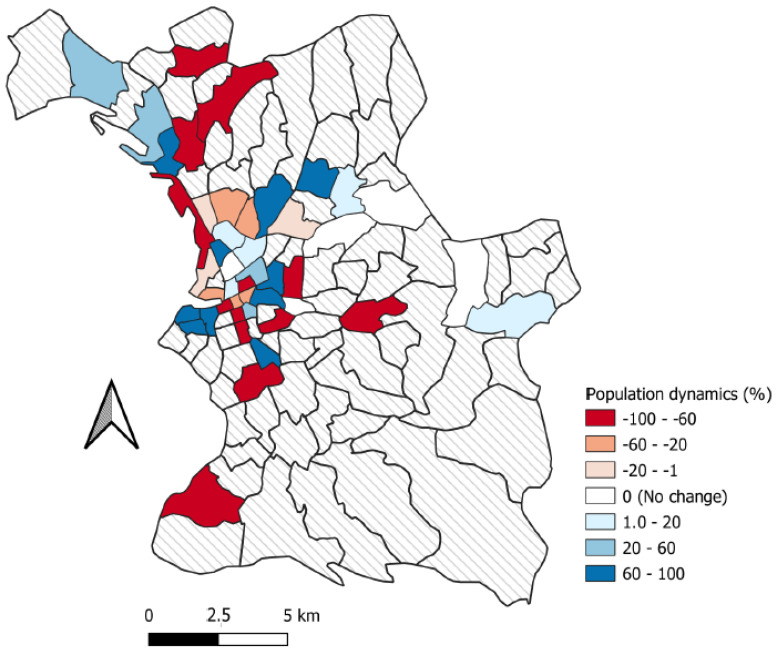
Population mobility between the first and second campaigns at the neighborhood scale. Neighborhoods in red indicate that they lose people between the first campaign and second campaign, neighborhoods in blue won people. Neighborhoods with no data are in hashed gray.

**Figure 6 ijerph-19-03129-f006:**
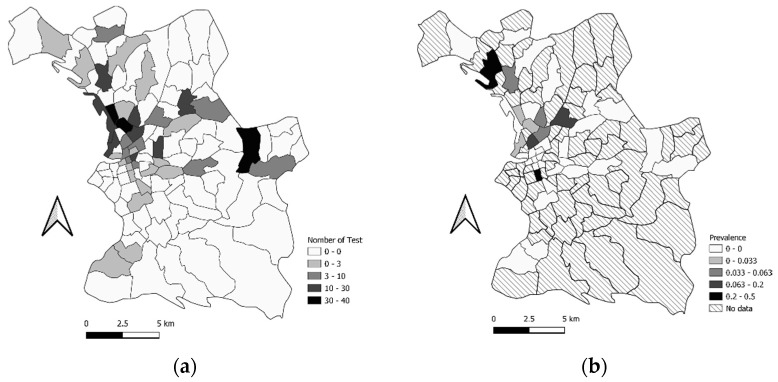
(**a**) Map of the number of tests that were performed by neighborhood for the first testing period (first campaign) in Marseille City, categorized into Jenks Natural Breaks Classification.; (**b**) Map of prevalence by the neighborhood for the first testing period (first campaign) in Marseille City, categorized into Jenks Natural Breaks Classification.

**Figure 7 ijerph-19-03129-f007:**
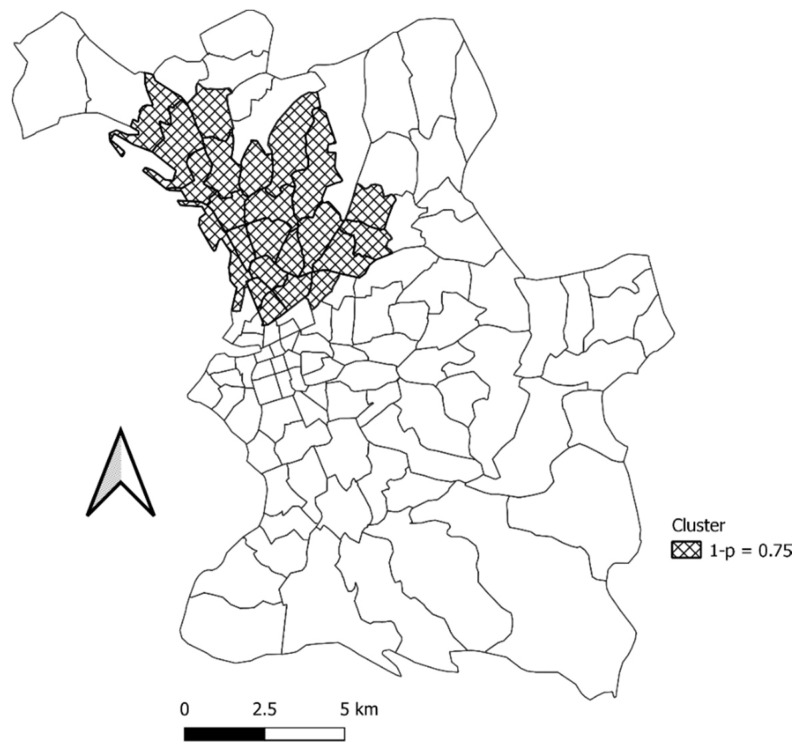
Map of the clusters that were identified for the first testing period (first campaign) in Marseille City.

**Figure 8 ijerph-19-03129-f008:**
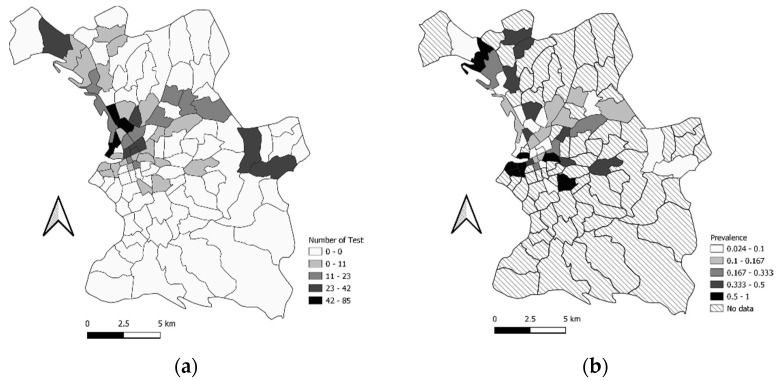
(**a**) Map of the number of tests that were performed by the neighborhood for the second testing campaign period in Marseille City, categorized into Jenks Natural Breaks Classification.; (**b**) Map of the prevalence by neighborhood for the second testing period in Marseille City, categorized into Jenks Natural Breaks Classification.

**Figure 9 ijerph-19-03129-f009:**
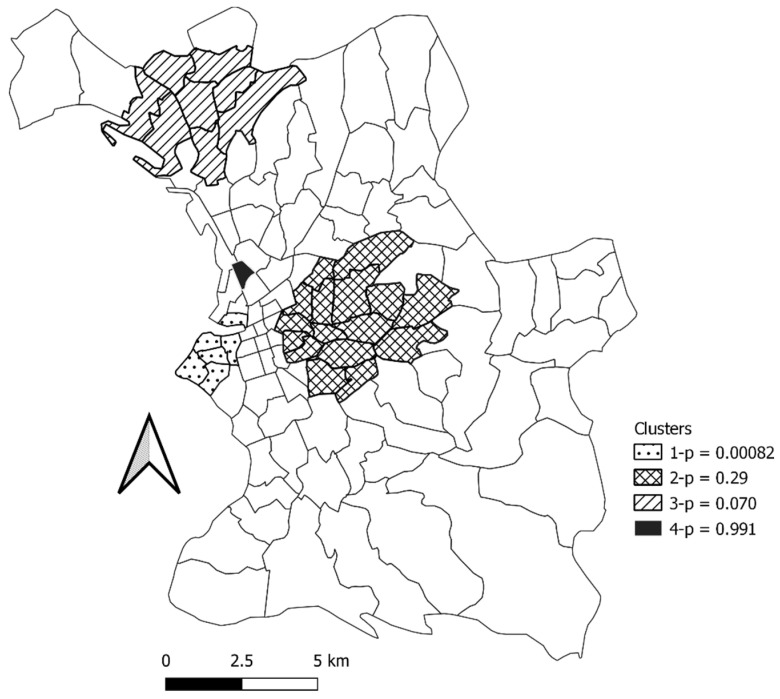
Map of the clusters that were identified in the second period testing period in Marseille City.

**Table 1 ijerph-19-03129-t001:** The sociodemographic characteristics of the study population who participated in the first campaigns (*n* = 1272).

Sociodemographic Characteristics	*n* (%) or Mean (SE)	
**Gender**		
Men	894	(70.29%)
Women	378	(29.71%)
Age (years)	40.06	(0.40)
Household status		
Isolated adult	672	(52.83%)
Family	416	(32.70%)
Isolated parent	130	(10.22%)
Missing	54	(4.25%)
Financial resources		
No	400	(31.45%)
Yes	794	(62.42%)
Missing	78	(6.13%)
Problems of economic resources during the period of health crisis		
No	321	(25.24%)
Yes	883	(69.42%)
Missing	68	(5.35%)
Country of Birth ^1^		
France	236	(18.55%)
European Union	199	(15.64%)
Europe, non-European Union	212	(16.67%)
North Africa	282	(22.17%)
Sub-Saharan/Southern Africa	213	(16.75%)
Middle East	15	(1.18%)
Russia	31	(2.44%)
North America	2	(0.16%)
South America	17	(1.34%)
Missing	65	(5.11%)
Education attainment		
No educational achievement	607	(47.72%)
Lower secondary	329	(25.86%)
Upper secondary or vocational	246	(19.34%)
Missing	90	(7.08%)
Health insurance		
No	247	(19.42%)
Yes	952	(79.84%)
Missing	73	(5.74%)
**Living Conditions**	* **n** * ** (%) or Mean (SE)**	
Total length of homelessness		
<3 months	90	(7.08%)
3 to 12 months	240	(18.87%)
1 to 5 years	452	(35.53%)
>5 years	397	(31.21%)
Missing	93	(7.31%)
ETHOS ^2^ Typology at baseline		
ETHOS 1: street	166	(13.05%)
ETHOS 2: emergency shelters and hotel rooms	447	(35.14%)
ETHOS 3: transitional shelters	172	(13.52%)
ETHOS 8: squats, slums	485	(38.13%)
Missing	2	(0.16%)
**Health Characteristics**	* **n** * ** (%) or Mean (SE)**	
Tobacco consumption		
No	486	(38.21%)
Yes	655	(51.49%)
Missing	131	(10.3%)
Alcohol consumption (glasses per day)	0.48	(0.03)
Substance consumption		
No	903	(70.99%)
Yes	218	(17.14%)
Missing	151	(11.87%)
Number of Comorbidities	0.57	(0.03)
Serological test for SARS-CoV-2		
Negative	1157	(90.96%)
Positive	74	(5.82%)
Missing	41	(3.22%)

^1^ “European Union” countries: Belgium, Bulgaria, Germany, Hungary, Italy, Poland, Portugal, Romania, Czech Republic, Slovakia, and Spain. “Outside European Union” countries: Albania, Armenia, Bosnia, Croatia, Moldavia, Montenegro, Serbia, Russia including Chechnya, and Ukraine. ^2^ ETHOS: the European typology for homelessness and housing exclusion.

**Table 2 ijerph-19-03129-t002:** Univariate analysis of the seroprevalence of SARS-CoV-2 infection between June and August 2020 in homeless people that were living in Marseille.

	HR (IC95%)	*p*-Value
Number of different accommodations in the past year	1.2 (1.007–1.424)	0.049

**Table 3 ijerph-19-03129-t003:** Results of the cluster analysis for the second period testing period, the neighborhoods concerned for each cluster are indicated in Appendix B Table A2.

Cluster	Population	Number of Cases	Expected Cases	Observed/Expected	Relative Risk	Log Likelihood Ratio	*p*-Value	Gini Cluster
1	5	5	0.51	9.88	10.53	11.608803	0.00082	yes
2	18	8	1.82	4.39	4.81	7.329902	0.029	yes
3	20	8	2.02	3.95	4.31	6.429304	0.070	no
4	5	2	0.51	3.95	4.03	1.552775	0.991	no

## Data Availability

The datasets that were generated and analyzed during the current study are not publicly available due to special authorization to transfer databases given by the CNIL. Upon prior authorization by the CNIL, the dataset would be available from the corresponding author on a reasonable request. Additionally, the study protocol is available for the request. All data requests should be directed to the corresponding author.

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
