# Peer review of "Residential Mobility of a Cohort of Homeless People in Times of Crisis: COVID-19 Pandemic in a European Metropolis"

_ijerph, 2022, doi:10.3390/ijerph19053129_

Round 1
Reviewer 1 Report
The authors should declare more clearly questions and aims of the article in the Introduction.
More information is needed about sampling: how homeless people have been identified and contacted? Did any selection occurred?
More information is needed about questions adopted in the questionnaire.
It is not clear the connection between questions and hypotheses of the research with scientific literature, which should be enlarged and better used.
It is not clear the function of figures 1 and 2 when they are presented: two lines which clarify how they will be used could be useful.
Is there any database to compare the sample composition with universe of homeless composition considering socio-demographic variables? If yes, a description of the sample distinctive traits would be very useful.
Similarly, the article refers to comparisons between homeless population and general population but it is not clear neither which data have been considered about general population, nor if the data of the present research are considered as representative of the “homeless population” on the whole and upon which bases this representativeness is justified
The authors must improve dialogue with literature in conclusions, highlighting possible unexpected results which emerged in the research.
The abstract refers to homeless policies but the topic seems not to be considered neither in the article nor in the conclusions.
Please check incomplete references: e.g. n. 16.
Author Response
The authors should declare more clearly questions and aims of the article in the Introduction.
Thank you for your comment, we have expanded the introduction in this sense.
Thus sentences were added or modified:
“We hypothesised that the mobility of homeless people would be disrupted by the SARS-Cov-2 epidemic in terms of housing type and between neighbourhoods in the city. We hypothesised that this mobility would differ according to the socio-economic characteristics of the homeless and that this mobility would have an impact in terms of epidemiology.” (Page 2 lines 42-46)
And
“In this study, we aimed to assess the homeless mobility and its epidemiological consequences in the context of the COVID-19 crisis.” (Page 3 lines 1-2)
More information is needed about sampling: how homeless people have been identified and contacted? Did any selection occurred?
More information is needed about questions adopted in the questionnaire.
Response: Thank you for your suggestion we have added the following paragraph to the manuscript:
“Participants were interviewed by trained cultural community health workers in the language of participants and using a questionnaire. This questionnaire collected data on sociodemographic characteristics, medical history and type and history of housing. The homes of participants were georeferenced." (Page 6 lines 10-13).
It is not clear the connection between questions and hypotheses of the research with scientific literature, which should be enlarged and better used.
Response: Thank you for your suggestion, we have completely rewritten the introduction to take this into account.
Thus sentences were added or modified:
“A study of homeless people in the Netherlands indicates that mobility and access to shelters opened at the beginning of the pandemic differed according to the category of origin of precarious people (Donker 2021). But we do not know what socio-economic variables are associated with them and what the exact consequences are for the mobility and sheltering of these people.
The pandemic is a good example of the effectiveness of the traditional type of rehousing programme in times of crisis. It allows us to see how the pandemic accelerated the problems of access to housing in various vulnerable populations.
We hypothesised that the mobility of homeless people would be disrupted by the SARS-Cov-2 epidemic in terms of housing type and between neighbourhoods in the city. We hypothesised that this mobility would differ according to the socio-economic characteristics of the homeless and that this mobility would have an impact in terms of epidemiology.” (Page 2 lines 34-46).
It is not clear the function of figures 1 and 2 when they are presented: two lines which clarify how they will be used could be useful.
Response: We have added the following information in 2.2 Study Area:
For figure 1:
This map shows the study area and is useful for associating district names with their locations in spatial analysis results. (Page 4 lines 1-2).
For figure 2:
Figure 2 shows the heterogeneous distribution of emergency accommodation in Marseille in relation to the different types of services. Figure 2, in the context of a study on precariousness and mobility, presents the state of health care supply and potentially the geographical factors of health inequality. (Page 4 lines 12-15).
Is there any database to compare the sample composition with universe of homeless composition considering socio-demographic variables? If yes, a description of the sample distinctive traits would be very useful.
Similarly, the article refers to comparisons between homeless population and general population but it is not clear neither which data have been considered about general population, nor if the data of the present research are considered as representative of the “homeless population” on the whole and upon which bases this representativeness is justified
Response: We have answered the question above regarding the representativeness of homeless people in our study. Where we compare our results to the general population, in the discussion, we have added the relevant references in the manuscript : (Insee 2021). (Page 16 line 27).
The authors must improve dialogue with literature in conclusions, highlighting possible unexpected results which emerged in the research.
Response: Thank you for your comment, we have completed the discussion and conclusion to better highlight the important results of our study.
Thus sentences were added or modified:
“In this paper we have shown that the most mobile individuals, with a greater number of different dwellings in the past year, were at greater risk of testing positive for SARS-CoV-2. The association between mobility and infection has already been demonstrated in the general population (Badr, Du et al. 2020, Nouvellet, Bhatia et al. 2021) but this is the first time it has been observed using the number of dwellings as an indicator in the homeless population.” (Page 17 lines 49-53).
“In summary, the mobility of homeless people at the city-scale is an important factor in better understanding the epidemiological dynamics for these populations. To date, these questions have been under investigated, despite concerning the public health of the most precarious people. Herein this study, we highlight the need for further research on these important issues. These results encourage the implementation of management adapted to the specific situations of these particularly vulnerable populations in times of health crisis. As this crisis can be seen as an accelerator of the usual sheltering movements, this study also allows us to reconsider the rehousing programme as a whole by advocating for housing first. This study is important at the city level for the adaptation of local strategies, but also at the European level for the implementation of more sustainable public housing programmes.” (Page 20 lines 23-33).
The abstract refers to homeless policies but the topic seems not to be considered neither in the article nor in the conclusions.
Response: Thank you for your comment, we have expanded the introduction and discussion to better highlight the role of homeless policies in our study and its conclusions.
Thus sentences were added or modified:
“To help homeless people exit homelessness, authorities implement rehousing programs. Most traditional rehousing programs assume that a person's homelessness is the result of poor decision making (Lyon‐Callo 2000, Manning and Greenwood 2019). Most traditional programs like “step-by-step” are clustered and supervised so that residents can become "housing ready" (Gulcur, Stefancic et al. 2003). Although the goal of rehousing programs is to end homelessness, on average it takes over 10 years for people to move out of these types of living conditions (Feantsa 2018). The reality is that many people are caught in an "institutional circuit" of homelessness, hospitals, and prisons (Hopper, Jost et al. 1997). There is growing evidence that, while traditional housing programs manage homelessness, they are not particularly effective in ending it (Sahlin 2005, Pleace 2008).” (Page 2 lines 3-11)
“We have shown that the number of dwellings is associated with a higher probability of infection. Going to a shelter is also associated with an increased likelihood of infection (Yoon, Montgomery et al. 2020, Baggett and Gaeta 2021, Mosnier, Loubière et al. 2021), especially if the shelters were overcrowded or the sleeping spaces shared with a large number of residents (Ghinai, Davis et al. 2020, Roederer, Mollo et al. 2021, Rogers, Link et al. 2021). Both of these factors argue for greater stability of people in times of health crisis in non-overcrowded living spaces, and approaches such as housing first seem to be more appropriate to address these issues.” (Page 18 lines 9-14).
Please check incomplete references: e.g. n. 16.
Response: Thank you for your comment, we have completed this reference as well as references 12,19,16,26.
Reviewer 2 Report
Typo and grammatical mistakes throughout the manuscript

Author Response
The study examines factors associated with COVID-19 infection among homeless people in France. While the population is understudied and a paper on people suffering from
homelessness and COVID-19 is an important public health issues, the authors need to
significantly revise the paper before it can be published. The following are the major issues I have spotted:
Outcomes and Data Analysis:
This section is quite confusing. The authors shall separate this part into two sections.
For example:
o All of the statistical analyses were carried out using R software [18], and
differences with p values of <0.05 were con-sidered statistically significant.
o Multivariate statistics were performed with the R package ade4 [21, 22].
This analysis generalizes the PCA (Principal Components Analysis) method to
be used with quantitative variables and factors [23]. The results and graphs
read like those of a PCA [24].
o “To find out if the number of accommodations in the past year was
significantly associated with having a positive serological test for SARS-Cov-2,
we used a multivariate logistic regression model.”
Response: Indeed, there was an error in the text that caused confusion: we did not make a multivariate logistic regression model but a univariate logistic regression model. Multivariate analysis refers exclusively to PCA. We have corrected this information in the text.
This sentence was modified:
“To find out if the number of accommodations in the past year was significantly associated with having a positive serological test for SARS-Cov-2, we used a univariate logistic regression model” (Page 7 lines 1-3).
- Explanatory variable should be written as independent variable or Predictors
Response: Thank you for your suggestion we have corrected in the text.
(Page 7 line 10).
- Response variable should be written as dependent or outcome variable.
Response: Thank you for your suggestion we have corrected in the text.
(Page 7 lines 11).
- Additional information is needed on how “rest” is measured and the range of fit
Response: detail added to Materials and Methods and Results sections 2.6.2. and 3.2.
Thus sentences were added or modified:
“To know the number of accommodations since the pandemic, we used the following question: “What is the person's current housing?” followed by the question “and before?”. We repeated the "and before?" question until we went back to the beginning of the pandemic. The total number of different housing units filled in corresponded to the number of accommodations since the pandemic.”
(Page 7, lines 12-16).
“The average number of different accommodations since the pandemic is 1.718 (standard error: 0.034), with 94 missing data. The number of different accommodations since the pandemic is significantly associated with having a positive serological test for SARS-Cov-2 (Table 2).”
(Page 10 lines 14-17).
- The PCA section is unclear, what factors did you use in the PCA analysis? Also, this seems more like latent class analysis to me
Response: We use the factor of the Table 1 we added this information in the text. It’s not latent class analysis, for more information see:
Hill, M. O., and A. J. E. Smith. 1976. Principal component analysis of taxonomic data with multi-state discrete characters. Taxon, 25, 249-255.
This sentence was added:
“We use the variables of the Table 1 to performed this analysis. We use stochastic regression imputation to asses variables for individuals with missing data, using the R package ‘mice’ (Van Buuren and Groothuis-Oudshoorn 2011).”
(Page 6 lines 44-16).
Results:
The result section should report findings of multivariate regression : What are the effect of SES factors, mobility on individual level and mobility on the spatial level, and prevalence of infection in own neighborhood on COVID-19 infection among homeless people in France. What I read in the results section is more about how SES, mobility on individual level and mobility on spatial level and prevalence of infection in neighborhoods are measured in your study. While these are important information and innovative methods on creating your predictors, the main analysis is missing.
Right now, the
Suggested format:
(1) Descriptive findings
(2) Measurement of predictors (i.e., 3.1 & 3.4)
(3) Bivariate findings (i.e., 3.2 & 3.3)
(4) Logistic regression findings
Response: There was an error in the text: we did not do a multivariate regression but a univariate regression. Multivariate analysis refers to Hill and Smith analysis, we have corrected this information in the text.
This sentence was modified:
“To find out if the number of accommodations in the past year was significantly associated with having a positive serological test for SARS-Cov-2, we used a univariate logistic regression model” (Page 7 lines 1-3).
To avoid confusion, we have systematically replaced in the text multivariate analysis by Hill and Smith analysis (Page 6 line 40, Page 9 line 3, Page 10 line 11, Page 21 lines 21-22). We welcome any comments on format changes now that this information is clear.
For section 3.1, so are there 4 types of people yielded based on the PCA using their SES
factors? The figure is not very informative, consider using another way to illustrates ...
also what SES factors you used to compose these four types of profiles?
Response: In this part we give the main axes of the PCA so the factors maximizing the best variance between individuals and then we indicate the variables most related to the 4 different types of ETHOS on the two axes of the analysis. This is the classic representation of this type of analysis. We don't really see how else to illustrate this but we are open to all suggestions.
Discussion :
- The author wrote :In this paper we have shown that the most mobile individuals,
with a greater number of different dwellings in the past year, were at greater risk of
testing positive for SARS-CoV-2.” If this is the main finding of this study, I do not see how the author reach this conclusion based on the data. Much more information is needed to demonstrate this conclusion.
Response: This statement is based on the analysis of the section "Relation between mobility at the individual scale and infection with SARS-Cov-2". We have completely rewritten the method and result parts of these analyses following your remarks.
Thus sentences were added or modified:
“To find out if the number of accommodations in the past year was significantly associated with having a positive serological test for SARS-Cov-2, we used a univariate logistic regression model. The independent variable was the number of accommodations since the pandemic and the dependent variable was the presence or absence of antibodies to SARS-CoV-2 at the individual level. To know the number of accommodations since the pandemic, we used the following question: “What is the person's current housing?” followed by the question “and before?”. We repeated the "and before?" question until we went back to the beginning of the pandemic. The total number of different housing units filled in corresponded to the number of accommodations since the pandemic.”
(Page 7 lines1-16)
“The average number of different accommodations since the pandemic is 1.718 (standard error: 0.034), with 94 missing data. The number of different accommodations since the pandemic is significantly associated with having a positive serological test for SARS-Cov-2 (Table 2).
Table 2. Univariate analysis of the seroprevalence of SARS-CoV-2 infection between June and August 2020 in homeless people living in Marseille.
|
. |
HR (IC95%) |
p-value |
|
Number of different accommodations in the past year |
1.2 (1.007-1.424) |
0.049 |
|
|
|
|
”
(Page 10 lines 14-20).
- Many new studies introduced in the discussion section which should be mentioned in the introduction section.
Response: Thank you for your suggestion, we have completely rewritten the introduction to take this into account.
Thus sentences were added or modified:
“The mobility of the homeless within large North American and European cities prior to the health crisis remains a little studied phenomenon (Snow and Mulcahy 2001). Some studies indicate that this mobility is greater than that of the general population in the same city (Gray, Chau et al. 2011, Tsai and Wilson 2020). Homeless people move within a city in order to find sustenance. According to Kaufman, 2020 (Kaufman 2020), research concerning the mobility of homeless people emphasizes moves within cities and reveals seven factors worthy of note: housing; labor markets; social, health, and justice services; personal health; the attributes of different places; interpersonal networks; and how mobility is socially differentiated. Homeless people from all kinds of accommodation were found to have a notable daily mobility (Šimon, Vašát et al. 2019), but little is known about their residential mobility over several months. Homeless people may also settle transiently, episodically or chronically in a shelter (Kuhn and Culhane 1998).
To help homeless people exit homelessness, authorities implement rehousing programs. Most traditional rehousing programs assume that a person's homelessness is the result of poor decision making (Lyon‐Callo 2000, Manning and Greenwood 2019). Most traditional programs like “step-by-step” are clustered and supervised so that residents can become "housing ready" (Gulcur, Stefancic et al. 2003). Although the goal of rehousing programs is to end homelessness, on average it takes over 10 years for people to move out of these types of living conditions (Feantsa 2018). The reality is that many people are caught in an "institutional circuit" of homelessness, hospitals, and prisons (Hopper, Jost et al. 1997). There is growing evidence that, while traditional housing programs manage homelessness, they are not particularly effective in ending it (Sahlin 2005, Pleace 2008).”
(Page 1 lines 37-45, Page lines 1-11).
- Much of the discussion is about the causes of homelessness, but not about factors associated with their high chances of getting infection. While such information is important, again, it has be oriented about the key findings of this study. Right now, it reads more like a methods paper, or more about a commentary.
Response: Thank you for your comment, we have completed the discussion and conclusion to better highlight the important results of our study.
This sentence was added:
“In this paper we have shown that the most mobile individuals, with a greater number of different dwellings in the past year, were at greater risk of testing positive for SARS-CoV-2. The association between mobility and infection has already been demonstrated in the general population (Badr, Du et al. 2020, Nouvellet, Bhatia et al. 2021) but this is the first time it has been observed using the number of dwellings as an indicator in the homeless population. This showed the importance of residential stability in order to comply with isolation and social distancing measures.”
(Page 17 lines 49-53, Page 18 lines 1-2).
- Rather than focusing the discussion on Marseille and why Marseille, I believe the
study will of greater significance if the authors can highlight that they are using
Marseille as a case here, and how the findings can be generalized and enhance our
understanding on the circumstances of homeless people on the international
context.
Response:
We have added the following information in the introduction:
“Marseille is a city of a rich country, a gateway to France and the European Union for many people from poorer countries. It could be representative of a city in a rich country close to poor countries attracting a high level of illegal and legal immigration.” (Page 2 lines 46-51).